# *Lobularia libyca*: Phytochemical Profiling, Antioxidant and Antimicrobial Activity Using In Vitro and In Silico Studies

**DOI:** 10.3390/molecules27123744

**Published:** 2022-06-10

**Authors:** Naima Benchikha, Imane Chelalba, Hanane Debbeche, Mohammed Messaoudi, Samir Begaa, Imane Larkem, Djilani Ghamem Amara, Abdelkrim Rebiai, Jesus Simal-Gandara, Barbara Sawicka, Maria Atanassova, Fadia S. Youssef

**Affiliations:** 1Chemistry Department, University of Hamma Lakhdar El-Oued, P.O. Box 789, El-Oued 39000, Algeria; naima_chem@yahoo.fr (N.B.); i.chelalbai@hotmail.com (I.C.); hanane-debbeche@univ-eloued.dz (H.D.); messaoudi2006@yahoo.fr (M.M.); djilani-ghemamamara@univ-eloued.dz (D.G.A.); 2Nuclear Research Centre of Birine, P.O. Box 180, Ain Oussera 17200, Algeria; samirbegaa@yahoo.fr; 3Agronomy Department, University of Mohamed Khider Biskra, P.O. Box 700, Biskra 07000, Algeria; imene.larkem@yahoo.com; 4Nutrition and Bromatology Group, Department of Analytical Chemistry and Food Science, Faculty of Food Science and Technology, University of Vigo—Ourense Campus, 32004 Ourense, Spain; jsimal@uvigo.es; 5Department of Plant Production Technology and Commodities Science, University of Life Science in Lublin, Akademicka 15 Str., 20-950 Lublin, Poland; barbara.sawicka@up.lublin.pl; 6Nutritional Scientific Consulting, Chemical Engineering, University of Chemical Technology and Metallurgy, 1734 Sofia, Bulgaria; 7Department of Pharmacognosy, Faculty of Pharmacy, Ain Shams University, Cairo 11566, Egypt; fadiayoussef@pharma.asu.edu.eg

**Keywords:** antioxidant activity, antibacterial activity, *L. libyca*, mineral content, molecular docking, total flavonoid, total phenol content

## Abstract

*Lobularia libyca* (*L. libyca*) is a traditional plant that is popular for its richness in phenolic compounds and flavonoids. The aim of this study was to comprehensively investigate the phytochemical profile by liquid chromatography, electrospray ionization and tandem mass spectrometry (LC-ESI-MS), the mineral contents and the biological properties of *L. libyca* methanol extract. *L. libyca* contains significant amounts of phenolic compounds and flavonoids. Thirteen compounds classified as flavonoids were identified. *L. libyca* is rich in nutrients such as Na, Fe and Ca. Moreover, the methanol extract of *L. libyca* showed significant antioxidant activity without cytotoxic activity on HCT116 cells (human colon cancer cell line) and HepG2 cells (human hepatoma), showing an inhibition zone of 13 mm in diameter. In silico studies showed that decanoic acid ethyl ester exhibited the best fit in β-lactamase and DNA gyrase active sites; meanwhile, oleic acid showed the best fit in reductase binding sites. Thus, it can be concluded that *L. libyca* can serve as a beneficial nutraceutical agent, owing to its significant antioxidant and antibacterial potential and due to its richness in iron, calcium and potassium, which are essential for maintaining a healthy lifestyle.

## 1. Introduction

Aromatic and medicinal plants are used all over the world, especially in China, India and many countries in Africa, in traditional medicine as a means of alleviating many ailments. The plant kingdom includes many medicinally useful plants that have been used by humans to produce new drugs as well as to treat various diseases. Recently, a large number of plants were screened for their possible pharmacological activities, among which are those belonging to the family *Brassicaceae* [1,2,3].

The family *Brassicaceae* consists of 338 genera and 3690 species, including 509 endemic species that are especially widespread in the Mediterranean region to Central Asia and Northwest America. The members of this family are often herbs and rarely shrubs or subtrees. The cabbage family (*Brassicaceae*) is known as the mustard family or cabbage, and it is also a large family with many plants of major economic importance. Due to their economic and agricultural importance, plants of the cabbage family have been and are still of considerable scientific interest, as they are the main dietary source of glucosinolates but are also high in minerals, vitamins, polyphenols, antioxidants and other bioactive compounds [4,5].

The *Brassicaceae* Burnett family plants have been widely used in the treatment of many diseases due to their anticancer, antibacterial, antifungal, antirheumatic and antidiabetic properties. Epidemiological studies have shown that the consumption of *Brassica* vegetables is strongly linked to the prevention of cardiovascular diseases and a decrease in cases of gastrointestinal cancers and other sites. Since antiquity, plants and products of the *Brassicaceae* Burnett family have been and are still of great interest to scientists thanks to their diverse biological activities, ranging from antimicrobial and antioxidant to anti-carcinogenic. The substances responsible for these properties are mainly phenolic compounds, glucosinolates and their derivatives [4,5].

*Lobularia libyca* (Viv.) CFW Meissner *(L. libyca*) (synonym: *Alyssum libyca* (Viv.)) belongs to the *Brassicaceae* Burnett family. This genus has five species native to the Mediterranean, Canary Islands and Cape Verde [6]. They are: *Lobularia arabica* (Boiss.) Muschl. (*L. arabica*), *Lobularia canariensis* (DC.) *L. Borgen* (*L. canariensis*), *L. libyca*, *Lobularia marginata* Webb ex Christ (*L. marginata*), and *Lobularia maritima* (L.) Desv. (*L. maritima*)—seaside Lobularia. They grow on dry sandy soils [6]. This species is an annual herbaceous desert plant native to North Africa in the Cape Verde Islands, Canary Islands and the Mediterranean region in the Arabian Desert. *L. libyca* is known as Khurm-El-ibra in Egypt; meanwhile, in Algeria, it is called Awaina Al-Hanash. It is an annual shrub up to 20–40 cm high. The plant is thin and completely covered with white hair, and its stem is erect and densely branched. It starts flowering in January and ends in March, while the fruiting period ends in May [6]. People have used *L. libyca* for many centuries as a culinary herb and herbal medicine. The plant is useful as an infusion to treat cough, cold and chest infections and in a syrup form for digestive upset and abdominal pains. It is also soothing for sore throat [7].

In silico molecular docking is considered an advanced protocol that can hasten the discovery of natural secondary metabolites with outstanding antihyperglycemic activity through virtual screening. It is defined as the use of multiple computational filters based on the diverse chemical structures present and collected in a well-known database with the aim to postulate the bioactivity of certain entities with respect to a specific target molecule. The main objective of virtual screening is to reduce the time, resources and efforts necessary for both in vitro and in vivo screening of defined compounds. Thus, by initial prediction of the inactive ones, the total number of compounds for further processing can be extensively reduced. Consequently, the hit scores in both in vitro and in vivo assays are dramatically elevated by excluding inactive compounds as compared to random assessment without preliminary virtual screening [8,9].

According to a comprehensive review of previous studies, nothing could be found in the literature regarding the biological or chemical properties of the plant growing in Algeria, except for its usage as an important source of grazing [6]. Hence, the main goal of this work was to study the phytochemical properties of the vegetable alcohol extract from *L. libyca*. These tests were performed using liquid chromatography coupled with mass spectrometry (LCMS) analysis and inductively coupled plasma (ICP-OES) for the determination of mineral contents, in addition to determining the total phenolic and flavonoid contents. The antioxidant activity was estimated using three methods: DPPH (2,2-Diphenyl-1-picrylhydrazyl), CAT (Catalase), and ABTS (2,2′-Azino-bis(3-ethylbenzothiazoline-6-sulfonic acid radical cations). The cytotoxic effect versus hepatic and colon cancer cell activity was also studied using the MTT (3-(4,5-Dimethylthiazol-2-yl)-2,5-Diphenyltetrazolium Bromide) assay. Antimicrobial activity against a panel of bacteria and fungi was assessed in vitro and further confirmed using in silico studies.

## 2. Materials and Methods

### 2.1. Plant Material and Extraction

The aerial parts of *L. libyca* (Figure 1) were collected in February 2018 from El-Oued province, Algeria. The identification of plant samples was confirmed by Dr. Djilani Ghamam Amara, lecturer at Hamma Lakhdar University of El-Oued, Algeria. The voucher specimen was carefully kept in El-Oued University, Algeria, in the herbarium, with code RO-030. The samples were washed with distilled water three times and were dried for four weeks, and then they were ground using an electric apparatus to obtain the dry matter mass, estimated at 50 g. Extraction using 80% methanol (methanol–water (8:2)) is a commonly used extraction technique that has been widely adopted in many research works [10,11,12,13]. Based on that, in our practical protocol, cold maceration was carried out using methanol–water (8:2, respectively) for 24 h and repeated three times.

The mixture was filtered and then dried under vacuum using a rotary evaporator at 45 °C to obtain the methanol extract.

### 2.2. Phytochemical Investigation of L. libyca Aerial Parts Using LC-ESI-MS Technique

*LC-ESI-MS* analysis was performed via injection of 100 μg/mL *L. libyca* methanol extract prepared from the aerial parts on Waters HPLC possessing a reverse-phase C-18 column (ACQUITY UPLC—BEH C18; 1.7 µm particle size—2.1 × 50 mm column). Elution was performed using a gradient, beginning with 10:90 (methanol–H_2_O) with 0.1% formic acid as a solvent system until 100% methanol using a flow rate of 0.2 mL/min, and the run lasted for 34 min. Meanwhile, mass spectrometry was performed by a XEVO TQD triple quadruple mass spectrometer. Negative ESI ionization ion mode was performed with the following conditions: Nitrogen was used as a nebulizing as well as a drying gas; 150 °C was used as the source temperature, while 440 °C was the desolvation temperature; capillary voltage and cone voltage were 3 kV and 30 eV, respectively. The cone gas flow and desolvation gas flow were 50 L/h and 900 L/h, respectively. Full scan mode was measured in the mass range of *m*/*z* 100–2000. The spectra were interpreted by Maslynx 4.1 software, and compounds were tentatively identified by comparing their mass spectra and retention times with what was previously reported in the literature [14,15].

### 2.3. Inductively Coupled Plasma ICP-OES

A scanning ICP-OES (inductively coupled plasma–optical emission spectrometer; Model: iCAP 7000 series) instrument with high resolution was used in the present work to determine the mineral element concentration in the extracts. All chemicals used in this investigation were of analytical grade (Sigma Aldrich, Germany). Three samples in powdered form of the studied plant weighing about 300 mg were placed into a Folin digestion tube. Six milliliters of the two-acid mixture (concentrated HNO_3_, HCl, 1:1) was added to the vessel and heated up at 105 °C until the solution became transparent. After 90 min, the tube was then filled up to 50 mL with distilled water to dilute the sample. Each sample was digested twice, and each diluted digestion solution was analyzed for mineral elements by ICP/OES twice. Then, all samples were stored at ambient temperature before mineral element analysis by the ICP/OES technique [16].

### 2.4. Determination of the Total Phenol and Flavonoid Contents

The total phenolic compounds were identified by spectrophotometry using the Singleton–Rossi method with the Folin–Ciocalteu reagent. Gallic acid was used as the reference of phenol at a wavelength (λ) = 765 nm. Gallic acid solutions diluted in methanol with known concentrations (mg/mL) were prepared to construct the standard curve. For the determination of the phenols in the plant extract, 1 mL of the extract was mixed with 0.5 mL of Folin–Ciocalteu reagent that was diluted 10 times with distilled water; then, this mixture was left for 5 min in the dark. To the latter, 2 mL of sodium carbonate (7.5%) was added, and the mixture was stirred in the tube, which was stored in the dark at laboratory temperature for 30 min. At the end, the absorbance of the solution obtained was read at a wavelength (λ) = 765 nm. To quantify the flavonoids in the plant extract, 1 mL of the plant extract was mixed with 1 mL of aluminum trichloride (AlCl_3_, 2%). The tube was shaken well and then left for an hour in the dark until the color turned yellow [15,17].

### 2.5. Determination of the Antioxidant Activity

#### 2.5.1. 2,2-Diphenyl-1-picrylhydrazyl Radical Scavenging Capacity Assay

To determine the rate of inhibition of free radicals (DPPH), ascorbic acid was used to compare it with the extract of the studied plant. For this, 1 mL of the prepared concentrations of the studied plant extract was placed in photovoltaic cells. To this volume, 200 μL of methanol and 800 μL of DPPH solution at a concentration of 0.4 mmol (4 mg of DPPH was dissolved in 100 mL of methanol) were added. Then, the samples were incubated in the dark for 30 min, and the optical density was read at a wavelength of 517 nm on a UV–visible device. The rate of inhibition of free radicals (DPPH) for the methanol extract of the studied plant was calculated from the obtained absorbance values and with mathematical calculations using the following relationship [15,18]: I% = [(A_0_ − A)\A_0_] × 100

A_0_: absorbance of DPPH at λ = 517 nm;A: absorbance of DPPH in the presence of the sample after 30 min at λ = 517 nm;I%: antioxidant inhibition ratio.

#### 2.5.2. Determination of Catalase Antioxidant Activity

The total antioxidant capacity of the methanol extract was estimated based on the method of phosphomolybdates, which acquires a light green color after the reduction of the molybdates. Gallic acid was used as the standard for the construction of a standard curve. A 0.1 mL sample of the extract was mixed with 1 mL of molybdate solution, stirred well and then left to stand for one hour in a water bath at 95 °C. After cooling the tubes, the optical absorption intensity was read at a wavelength of 695 nm on the UV-VIS spectrophotometer [16,19].

#### 2.5.3. 2,2’-Azino-bis(3-ethylbenzothiazoline-6-sulfonic Acid Radical Cation Scavenging Assay

BHT (butylhydroxytoluene) was used as the standard for the construction of a standard curve. First, 1 mL of ABTS solution was mixed with 50 µL of plant extract in test tubes, which were shaken and then left in the dark for 10 to 30 min at room temperature (30 °C). Next, the absorbance was read at 734 nm wavelength using the UV-VIS spectrophotometer. The inhibition rate (I%) of the extracts was measured by the following relationship [20].
 I% = [(A_0_ − A)\A_0_] × 100

A_0_: absorbance in the absence of the inhibitor (control);A: absorbance in the presence of the inhibitor (sample);I%: inhibition rate.

### 2.6. Assessment of the Cytotoxic Activity In Vitro by MTT Assay

The cytotoxic activity of the plant extract was tested against both HCT116 (colon cell line (ATCC^®^ CCL247™)) and HepG2 (human hepatocellular carcinoma cell line (ATCC^®^ HB-8065™)) using the mitochondrial-dependent reduction of yellow MTT (3-(4,5-dimethylthiazol-2-yl)-2,5-diphenyl tetrazolium bromide) to purple formazan [21]. The in vitro bioassay on human tumor cell lines was conducted and determined by the Bioassay-Cell Culture Laboratory, National Research Centre, El-Tahrir St., Dokki, Cairo 12622, Egypt.

### 2.7. Assessment of the Antimicrobial Activity In Vitro

#### 2.7.1. Microbial Strains

*L. libyca* extract was tested against a panel of microorganisms, including standard Gram-positive and Gram-negative bacteria and yeasts as well as fungi, which were obtained from the Pasteur Institute in Algeria. These include *Escherichia coli* (ATCC25922), *Staphylococcus aureus* (ATCC2592), *Klebsiella pneumonia* (ATCC700603), *L. innocua* (CLIP74915) and *Candida Albicans* (IPA200).

#### 2.7.2. Determination of Mean Inhibition Zones

After the growth of fungi and bacteria in culture media for ten days, the surface of the bacteria, as well as that of the fungi and yeasts responsible for plant diseases, was scraped with an L-shaped glass rod and then dissolved in 5 mL of sterile physiological water. The working surface was sterilized with a benzene burner, and then the nutrient solution was poured into Petri dishes. It was dissolved in a Muller–Hinton (MH) agar sterilizer for bacteria, and the dishes were inoculated with the medium (*Sabouraud* dextrose agar) (SDA) with 100 μL of the solution prepared for fungi or bacteria in the presence of a benzene burner on the worktable; then, the sterilized dishes were cooled and frozen. The extract was dissolved in methanol to obtain a concentration of 20 mg/mL for fungi, while for bacteria, an extract with a concentration of 10 mg/mL was prepared. To grow bacteria and fungi, a cotton swab was dipped in the bacterial or fungal suspension and then wiped over all of the previously prepared culture media in *Petri* dishes in the form of converging lines, repeating the process 3 times and rotating the plate at an angle of 60° each time. Then, 10 μL of the extract was taken by pipette and placed on sterile discs of filter paper (6 mm in diameter), and then they were put on the surfaces of agar inoculated with fungi or bacteria; methanol was used as a negative control. The finished dishes were placed at 4 °C for 2 h, and the Petri dishes were placed upside down in a 37 °C incubator for 24 h for bacteria and 72 h for fungi. An antibiotic was used to compare the biological effects of the plant extract to that of the antibiotic used, namely*, Amoxyclav* AMC30:30 µg/disc. Once the incubation period was over, the *Petri* dishes were taken to measure the inhibition zone diameters (mm) of the extracts, the antibiotic and the control, which reveal the antimicrobial activity of the extract [22].

### 2.8. In Silico Studies

#### 2.8.1. Validation of Molecular Docking Studies Using Re-Docking and Superimposition

Validation of molecular docking was performed for all of the executed experiments by comparing the alignments of the highly stable docking poses of the lead compound with the lead conformer co-crystallized with the respective enzyme from pdb. The RMSD (Root Mean Square Deviation) value was used to confirm the validity of the docking experiment, which in turn reflects the ability to predict the binding affinity of new ligands [23,24].

#### 2.8.2. Molecular Docking Studies

In silico molecular modeling experiments were carried out for the major compounds identified in *L. libyca* extract within the active sites involved in the incidence of bacterial infection and antibiotic resistance development. These are β-lactamase (PDB ID 3NBL; 2.0 A°) [25], DNA-gyrase (PDB ID 4Z2D; 3.38 A°) [26] and dihydrofolate reductase (PDB ID 4KM2; 1.4 A°) [27], and Discovery Studio 2.5 (Accelrys Inc., San Diego, CA, USA) was used, applying the C-Docker protocol as previously described [28]. The X-ray crystal structures of the used enzymes were obtained from the Protein Data Bank in PDB format. Preparation of the structure of the used enzyme structure was performed by applying the default protocol for protein preparation in Discovery Studio 4.5 (Accelrys Inc., San Diego, CA, USA). This includes the removal of water molecules and the addition of hydrogen atoms, which are consequently followed by freeing the protein structure from any unwanted interactions. CHARMm was selected as the forcefield, whereas MMFF94 was adopted as a method for the calculation of partial charge, accompanied by minimization of the added hydrogen in about 2000 steps. Determination of the binding center was achieved on the basis of the reported data addressing the enzyme catalytic domain. The 2D structures of the tested compounds were drawn in ChemDraw 13.0 and saved in the form of pdb files. Preparation of the structures was carried out by employing the default protocol for ligand preparation in Discovery Studio 4.5 (Accelrys Inc., San Diego, CA, USA). Consequently, docking of the prepared compounds was carried out within the active sites of the energy-minimized protein by using the C-Docker protocol and leaving its parameters as its default parameters. The CHARMm force field was assigned, and the binding energies were calculated using the distance-dependent dielectric implicit solvation model for the selected docking poses.

The free binding energies (∆G) were calculated in Kcal/mol using the following equation:ΔG_binding_ = E_complex_ − (E_protein_ + E_ligand_)(1)
where:ΔG_binding_: the ligand–protein interaction binding energy;E_complex_: the potential energy for the complex of protein bound with the ligand;E_protein_: the potential energy of protein alone;E_ligand_: the potential energy for the ligand alone.

### 2.9. Statistical Analyses

All data measurements are presented as mean ± standard deviation (SD) and were analyzed by one-way analysis of variance (ANOVA), followed by Tukey’s multiple range tests; *p* < 0.05 was considered the level of significance (n = 3). The calculations were made with the SAS Enterprise 4.2 software. The descriptive statistics of the studied traits were analyzed using the IBM program SPSS Statistics 26 [29,30]

## 3. Results and Discussion

### 3.1. Phytochemical Investigation of L. libyca Aerial Parts Using LC-ESI-MS Technique

Phytochemical investigation of the total methanol extract of *L. libyca* aerial parts using LC-ESI-MS analysis (Figure 2) resulted in the tentative identification of thirteen compounds belonging mainly to the flavonoid family. These compounds are represented by kaempferol 3-O-trihexoside 7-O-α-L-rhamnopyranoside, kaempferol 3-O-dihexoside 7-O-α-L-rhamnopyranoside [6], kaempferol 3-O-hexosyl- α- rhamnopyranoside-7-O-α-rhamnopyranoside [31], kaempferol 3-O-trihydroxybenzoyl- α-L-arabinofuranoside [32], kaempferol 3-O-hexoside, 7-O-[4-hydroxy-3,5-dimethoxy-E-cinnamoyl dihexoside [33], kaempferol 3-O-[4-hydroxy-3,5-dimethoxy-E-cinnamoyl dihexoside 7-O-α-L-rhamnopyranoside [34], apigenin 7-O-[3,4-di-O-acetyl-α-L-rhamnopyranosyl-hexoside [35], kaempferol 3,7-di-O-α-rhamnopyranoside [31], tetrahydroxyflavanone- trihydroxyflavone [36] and hexahydroxyflavan, 3-O-trihydroxybenzoyl [37]. In addition, phenolic acid as quinic acid [38] and fatty acids represented by oleic acid [39] and decanoic acid ethyl ester [40] were identified. A scheme representing some of the major identified compounds is illustrated in Figure 3.

These results indicate the richness of *L. libyca* in flavonoids, which account for 49% of the total identified compounds, which is consistent with what was previously isolated from *L. libyca* [1]. Compounds identified in *L. libyca* are illustrated in Table 1.

### 3.2. Mineral Elemental Analysis

#### Mineral Elemental Analysis

The mean values of mineral elemental concentrations evaluated in *L. libyca* are presented in Table 2. The mineral concentration was determined based upon the dry weight of the plant samples by taking the mean value of three replicates and is represented as mean ± SD. In this study, some minerals were detected in the leaves of *L. libyca* growing in Algeria, where thirteen mineral elements were determined by the ICP technique, namely: P, Na, K, Mg, P, S, Fe, Cu, Zn, Mn, B, Mo and N, representing eight essential elements, which are Na, K, Ca, Fe, Mn, Zn, Mg and K, and five non-essential elements [41,42,43]. Many types of research and pharmaceutical applications of medicinal plants, whether on extracts or volatile oils, have been conducted by many researchers around the world, resulting in great progress in treating various diseases. However, the absorption of minerals by herbs and plants, whether nutritional or toxic, varies highly in plants with the level of heavy metals in the soil, atmosphere, irrigation and water [44,45,46].

The results of these studies showed the large diversity of the tested extract in terms of mineral compounds. The coefficient of variation, which is a measure of suppuration, is used to test the degree of variation in the value of a variable. A high value of the coefficient indicates the large differentiation of the feature and proves the heterogeneity of the tested feature, whereas a low value proves the low variability of the feature and the homogeneity of the examined elements. Among the analyzed macro- and microelements, molybdenum (V = 2.47%) turned out to be the most stable, while phosphorus (V = 24.06%) and calcium (V = 16.80%) turned out to be the most variable. A similar tendency was observed in the case of mineral profiling of other plant species assessed in Morocco [47]. The results of these studies showed the large diversity of the tested extract in terms of mineral compounds. The coefficient of variation, which is a measure of suppuration, is used to test the degree of variation in the value of a variable. A high value of the coefficient indicates the large differentiation of the feature and proves the heterogeneity of the tested feature, whereas a low value proves the low variability of the feature and homogeneity of the examined elements. Among the analyzed macro- and microelements, molybdenum (V = 2.47%) turned out to be the most stable, while phosphorus (V = 24.06%) and calcium (V = 16.80%) turned out to be the most variable (Table 2). A similar tendency was observed in the case of mineral profiling of other plant species assessed in Morocco [47].

The uptake of metals by plants strongly depends on several factors, such as the plant morphology, chemical and physical forms of the adsorbed metal, the duration of exposure, environmental conditions, the concentrations of pollutants and other natural factors. All of these factors significantly influence the qualities and the concentration of the metals in the plant [48]. Thus, it is important to estimate the mineral concentrations in the studied plant as well as to compare them with the permissible levels recommended by the World Health Organization (WHO 2021) [49,50].

It has been shown that the tested plant species is a useful source of minerals. Important correlations were found between the various minerals. These results were confirmed by ANOVA and descriptive statistical analysis. According to Ibourki [47], the obtained results show that *L. libyca* may be a promising source of essential minerals in the human diet and in other applications.

Several previous studies reported that human health needs several elements that are considered micronutrients, including potassium and sodium, where both electrolytic elements K and Na are responsible for maintaining a normal fluid balance inside and outside cells [51,52,53]. Moreover, it is recommended that the concentration levels of K be major and Na be minor; in this study, the concentrations of K and Na were found to be 24,564 and 12,442 mg/kg, respectively. Macroelements such as calcium are indispensable for life, as Ca is crucial for the body’s growth and physiological functionality and is essential for healthy bones, teeth and blood. Calcium is also necessary for the absorption of dietary vitamin B, the synthesis of the neurotransmitter acetylcholine and the activation of enzymes such as pancreatic lipase [54,55]. The content of Ca was found to be 41,126 mg/kg. The concentration of Zn in *L. libyca* was quite significant and estimated to be 55 mg/kg. Zinc can exert protective effects on liver cells by inhibiting lipid peroxidation and stabilizing cell membranes; on the other hand, zinc deficiency may lead to diarrhea, weight loss and ulcer-healing problems, and these conditions may become fatal if not recognized and remain untreated [56].

Manganese plays an essential role in bone mineralization and cellular protection from damaging free radical species [53,57]. The content of Mn in the studied plant was equal to 14.06 mg/kg. Iron is a micronutrient for human health, playing an essential role in the body, and it is associated with hemoglobin and with the transfer of oxygen from the lungs to the tissue cells. Iron deficiency is usually caused by insufficient nutritional intake or multiple births in mothers, causing anemia; hence, the daily intake of iron is necessary [58,59]. The concentration of Fe was about 353 mg/kg in the sample analysis of the studied plant. Additionally, copper, an essential mineral for human health, has a significant effect, through which it enables the body to make red blood cells and may help to prevent cardiovascular disease and osteoporosis. Cu helps to maintain healthy blood vessels and immune function, and it contributes to iron absorption [60]. Cu exists in the range of 8.13 mg/kg in Algerian *L. libyca*.

Magnesium (Mg) is a mineral essential for many body functions, and it maintains nervous and muscular balance and prevents fatigue and stress. Magnesium participates in more than three hundred enzyme reactions and helps to maintain bone structure. It also contributes to muscle relaxation and nerve impulse transmission, to the production of proteins and to the regulation of the heart rate [45]. The concentration of Mg in *L. libyca* was quite significant and estimated at 6651 mg/kg. According to the results obtained, established in Table 2, it was concluded that Algerian *L. libyca* represents a potential source of nutrient elements.

### 3.3. Total Phenol Content and Flavonoids

The results obtained for the total contents of phenols and flavonoids are illustrated in Table 3. From the results, it is notable that the plant contained considerable quantities of phenolic compounds, as well as flavonoids. As plants of the Brassicaceae family are considered the main sources of phenolic compounds in human food, it was expected that there would be significant amounts of phenols and flavonoids in the *L. libyca* plant, which belongs to this family.

A comparison with an Egyptian study was made, as according to previous research, the latter is the only one that carried out analyses similar to what we performed on this plant. Otherwise, most other researchers have only looked at the plant from a morphological or regional point of view, as well as its use as a pastoral plant. The results of this comparison are summarized in Table 4. The Egyptian form had a significantly higher total content of flavonoids, while the *L. libyca* Algerian form had a higher content of phenols. At first glance, it should be noted that the two studies concluded that this plant contains significant amounts of phenols and flavonoids, but for many reasons, the results should be different between the Algerian and Egyptian plants. Differences in the climate and environment, as well as the period of plant picking, are all factors that lead to these divergent results.

### 3.4. Antioxidant Activity

Recently, natural foods and food-borne antioxidants such as vitamins and phenolic phytochemicals have received increasing attention, as they are known to act as chemical protective agents against damage caused by oxidative stress. It is well known that flavonoids and plant phenols are generally very effective in removing free radicals and are considered antioxidants. Polyphenols and flavonoids are used to prevent and treat various diseases, mainly those related to free radicals. Thus, due to the richness of this plant in phenols and flavonoids, the measurement of the antioxidant activity of *L. libyca* was carried out using three methods, namely, (DPPH (2,2-Diphenyl-1-picrylhydrazyl), CAT (Catalase) and ABTS (2,2′-Azino-bis(3-ethylbenzothiazoline-6-sulfonic acid radical cations). The results are summarized in Table 5. By comparing the three methods, it was found that the best percentage of inhibition was that with the DPPH inhibition method. From the methanol extract activity curve of the inhibition of DPPH free radicals, as well as the standard curve of ascorbic acid, the IC50 value of the extract was calculated, knowing that it is the lowest value of the latter, which represents the best inhibitory effect. The results showed that the alcoholic extract of the plant displayed antioxidant activity using the free radicals of DPPH, and the activity was compared to ascorbic acid. This inhibition was somewhat weak compared to the value of the latter (ascorbic acid). According to previous studies, the results of inhibition by total alcohol extract were not released due to its weak effect, and this is consistent with our results. These studies showed that the highest inhibition rate was obtained with butanol extract, and this will be the subject of our future study. The antioxidant activity and the total phenol and flavonoid contents of the *L. libyca* plant are positively correlated, which indicates that the phenolic compounds contribute to the antioxidant activity, confirming what we previously mentioned. The antioxidant activity of *L. libyca* was 77.33 in mg TE/g of extract. The APTS value was 95.25 ABTS (mg TE/g extract), and the CAT value was 180.06 mg AG/g extract. The IC50 value of *L. libyca*, determined by the DPPH test, was 357.62 μg/mL.

### 3.5. Anticancer Activity

The results of the cytotoxicity assessment illustrated in Table 5 show that there was no response or a lack of efficacy of the methanol extract of the aerial parts of this plant against liver and colon cancer cells. These tests on these types of cancer cells, which were carried out at the National Centre in Egypt, are the first of their kind on this plant, because no previous studies were detected either in Algeria or in other countries to compare and verify them. As mentioned before, research on the seeds of this plant in Egypt has shown that it is effective against cancer cells in the liver, colon and breast, while it is not effective against cancer cells in the lung (Table 5).

### 3.6. Antimicrobial Activity

The antibacterial activity of *L. libyca* extract was determined in vitro using the agar well diffusion method versus different standard Gram-positive and Gram-negative bacteria by assessing the mean inhibition zones, as illustrated in Table 6. *L. libyca* extract showed significant antibacterial potential against the tested Gram-positive and Gram-negative bacterial strains, with mean inhibition zone diameters ranging between 11–13 mm, approaching that of the tested antibiotic, which showed 23 mm as the diameter of zone inhibition. *E. coli* was the most susceptible to the extract’s antimicrobial effect, showing a mean inhibition zone diameter of 13 mm, followed by S. *aureus* and *K. pneumonia*, displaying a mean inhibition zone diameter of 13 mm. Meanwhile, *L. innocua* was the least susceptible, with a mean inhibition zone diameter equal to 11 mm. Furthermore, the antifungal activity of *L. libyca* extract was assessed in vitro using *Sabourad* dextrose agar plates. Unfortunately, these fungi exhibited low sensitivity to the tested extracts, where the diameters of inhibition varied between 0.9–1.3 mm (Table 6). Plants contain many compounds that possess antibacterial activity, the most important of which are phenolics, flavonoids and terpenes [61], which are present in plant extracts, and consequently, positive results were expected. The effect of antibacterial activity can be explained by the absorption of phenolic compounds by bacterial cell membranes and their interaction with active enzymes and mineral elements in cell walls. The difference in sensitivity between bacterial strains (Gram-positive and Gram-negative) is due to differences in the structure, composition and nature of the bacterial cell wall, as well as the presence of lipopolysaccharides contained in the outer membrane of bacteria (Gram-negative) [62,63].

### 3.7. In Silico Studies

#### 3.7.1. Validation of Molecular Docking Studies Using Re-Docking and Superimposition

Validation experiments revealed a good alignment between the docking pose of the lead compounds that exhibited the best fit with the lead conformers that co-crystallized with their respective enzymes. They showed RMSD values of 0.84, 1.11 and 1.83 Å for β-lactamase, DNA-gyrase and dihydrofolate reductase, respectively. Thus, it greatly ensured the validity of the docking experiment since RMSD was less than 2 angstroms [23,24] (Figure 4).

#### 3.7.2. Molecular Docking Studies

In silico molecular docking was performed, aiming to find the exact mode of action of these compounds to combat microbial infections. It is worth highlighting that many enzymes could be a target, either to alleviate bacterial infections or to inhibit the appearance of bacterial resistance to popular drugs. DNA-gyrase is a crucial enzyme that controls DNA supercoiling, in addition to relieving the topological stress arising from the translocation of transcription and replication complexes within DNA. However, folic acid is very important to the growth and multiplication of bacteria [28]. Dihydrofolate reductase acts as a catalyst for the NADPH-dependent reduction of dihydrofolate to tetrahydrofolate, where the produced metabolites of tetrahydrofolate are mandatory for the incorporation of single carbon units into purines, pyrimidines and amino acids. Hence, the inhibition of dihydrofolate reductase causes a deficiency of the components of nucleic acids and proteins with the subsequent prohibition of DNA synthesis, which finally causes cell death, and thus, substances that inhibit it can effectively be used as antibacterial agents [26]. Concerning bacterial resistance, *β*-lactamases represent enzymes that are produced by bacteria and are crucial for the development of resistance to multiple antibiotics with *β*-lactam rings, such as penicillin, by cleaving the *β*-lactam ring, resulting in the destruction of antibiotic activity. Therefore, exploring *β*-lactamase inhibitors is critical for combating bacterial resistance [25]. Molecular docking was carried out for the major compounds identified in *L. libyca* extract within the active sites of crucial proteins involved in the growth and replication of microbes, as well as in causing their resistance. These are *β*-lactamase (PDB ID 3NBL; 2.0 A°), DNA-gyrase (PDB ID 4Z2D; 3.38 A°) and dihydrofolate reductase (PDB ID 4KM2; 1.4 A°). The results are displayed in Table 7 and Figure 3. Decanoic acid ethyl ester showed the best fit within the active sites of *β*-lactamase and DNA-gyrase, with free binding energies equal to −37.34 and −27.16 Kcal/mole, respectively. These values approach that of cefuroxime (∆G = −61.76 Kcal/mole in *β*-lactamase active site); meanwhile, decanoic acid ethyl ester showed superior activity towards DNA-gyrase when compared with levofloxacin (∆G = −9.90 Kcal/mole). However, oleic acid displayed the best fit within dihydrofolate reductase binding sites, revealing ∆G = −32.52 Kcal/mole, exceeding that of trimethoprim (∆G = −30.10 Kcal/mole) (Table 7). The tight binding of decanoic acid ethyl ester to *β*-lactamase can be interpreted by virtue of the formation of multiple bonds with the amino acid residues at the active site, including two conventional H-bonds with Lys87 and Ser84, one C-H bond with Gly144 and one alkyl bond with Leu 185, in addition to multiple Van der Waals bonds. Regarding DNA-gyrase, decanoic acid ethyl ester forms one C-H bond with Gly434, in addition to multiple Van der Waals bonds with the amino acid moieties at the active site. Concerning dihydrofolate reductase, the firm binding of oleic acid to the active sites is attributed to the formation of two H-bonds with Arg60 and Val54 and one C-H bond with Pro58, in addition to many Van der Waals interactions (Figure 5). Oleic acid was previously reported to exert antimicrobial activity, evidenced by killing group A streptococci via changing the integrity of the cell membrane with the concomitant loss of ribonucleic acid but not deoxyribonucleic acid [64]. In addition, decanoic acid and its derivatives were previously reported in many studies to possess considerable antibacterial and antifungal activities, and its esters have been employed in medical, nutritional and dietetic fields. It is potentially valuable for reducing the level of colonization of chicks and could ultimately aid in decreasing the number of contaminated eggs in the food supply (4). QSAR studies indicated that activity against Gram-positive bacteria was controlled by the lipophilicity of the compound, while the topological steric nature of the molecule was a determining factor for antifungal activity [24,65].

## 4. Conclusions

Herein, quantitative and qualitative analyses of the methanol extract of the aerial parts of *L. Libya* show that it contained large amounts of phenols and flavonoids, as well as exhibited antioxidant activity, as estimated by several methods. It showed considerable antimicrobial activity using both in vitro and in silico methods. The analysis of the minerals carried out on the dry plant showed that the plant contained several minerals important for living organisms, such as potassium, sodium, zinc, iron and phosphorus, either in their fresh form or as good soil fertilizer in case of death. Regarding the inhibitory capacity against liver and colon cancer cells, the plant was negative for both types. The results of the current work may be beneficial and can be used as a database for researchers and specialists and to enrich the medicinal herbs database. However, further in-depth in vivo studies are highly recommended to be conducted to further ascertain the obtained results. The mineral contents of *L. libyca* aerial parts are rich in essential nutrients such as Na, Fe, Ca and K. *L. libyca* methanol extract showed notable antioxidant activity in the 2,2-diphenyl-1-picrylhydrazyl assay (DPPH) (77.33 TE/g), catalase assay (CAT) (95.24 mg TE/g) and 2,2'-azino-bis(3-ethylbenzothiazoline-6-sulfonic acid (ABTS) assay (180.06 mg eq. AG/g), with no cytotoxic effect on HCT116 (human colon cancer cell line) and HepG2 (human hepatoma) cells. It showed considerable antimicrobial activity against standard Gram-positive and Gram-negative bacteria, where E. coli was the most susceptible to the extract’s antimicrobial effect, showing a mean inhibition zone diameter of 13 mm. This was supported by in silico studies, where decanoic acid ethyl ester showed the best fit within the active sites of β-lactamase and DNA-gyrase, with free binding energies equal to −37.34 and −27.16 Kcal/mole, respectively. Oleic acid displayed the best fit within dihydrofolate reductase binding sites, revealing ∆G = −32.52 Kcal/mole. Thus, *L. libyca* can serve as a beneficial nutraceutical agent, attributed to its considerable antioxidant and antimicrobial potential, as well as its richness in iron, calcium and potassium, which are crucial for maintaining a healthy life.

## Figures and Tables

**Figure 1 molecules-27-03744-f001:**
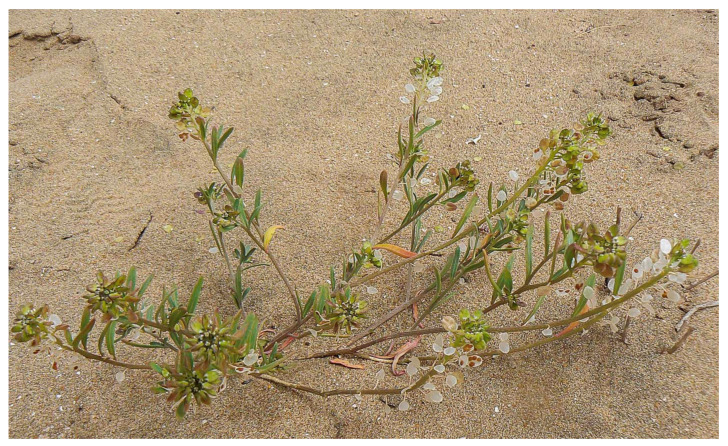
The aerial parts of *L. libyca* (Viv.) Meisnner plants (leaves and flowers) (source: Fouad Msanda).

**Figure 2 molecules-27-03744-f002:**
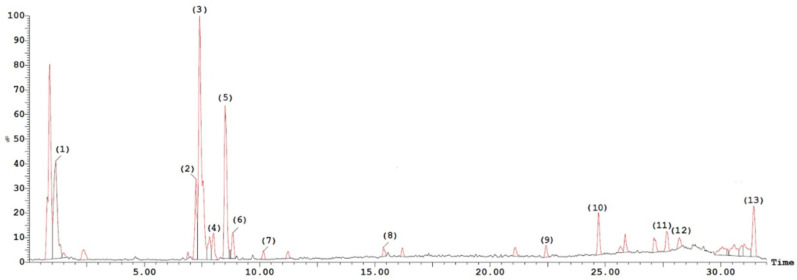
LC/MS chromatogram of the total methanol extract of *L. libyca* aerial parts in negative ionization mode.

**Figure 3 molecules-27-03744-f003:**
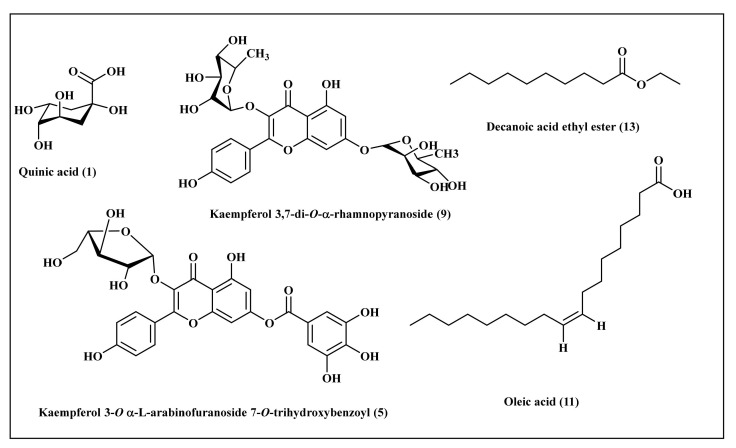
Scheme represents some of the major identified compounds from the methanol extract of the aerial parts of *L. libyca*.

**Figure 4 molecules-27-03744-f004:**
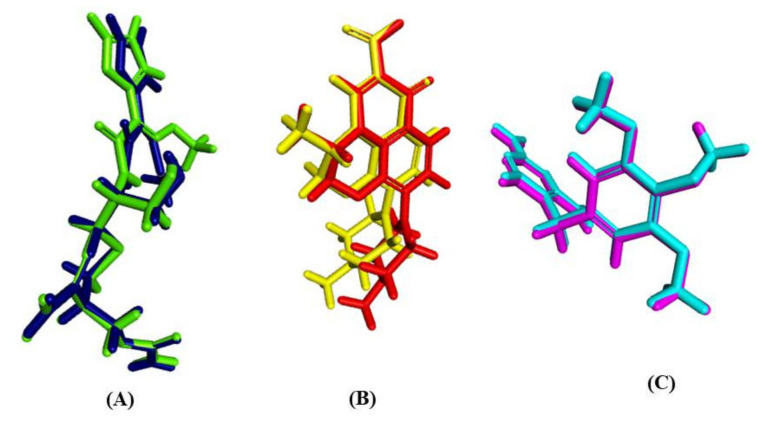
Validation of the docking experiments for *β*-lactamase (**A**), DNA-gyrase (**B**) and dihydrofolate reductase (**C**) using re-docking and superimposition.

**Figure 5 molecules-27-03744-f005:**
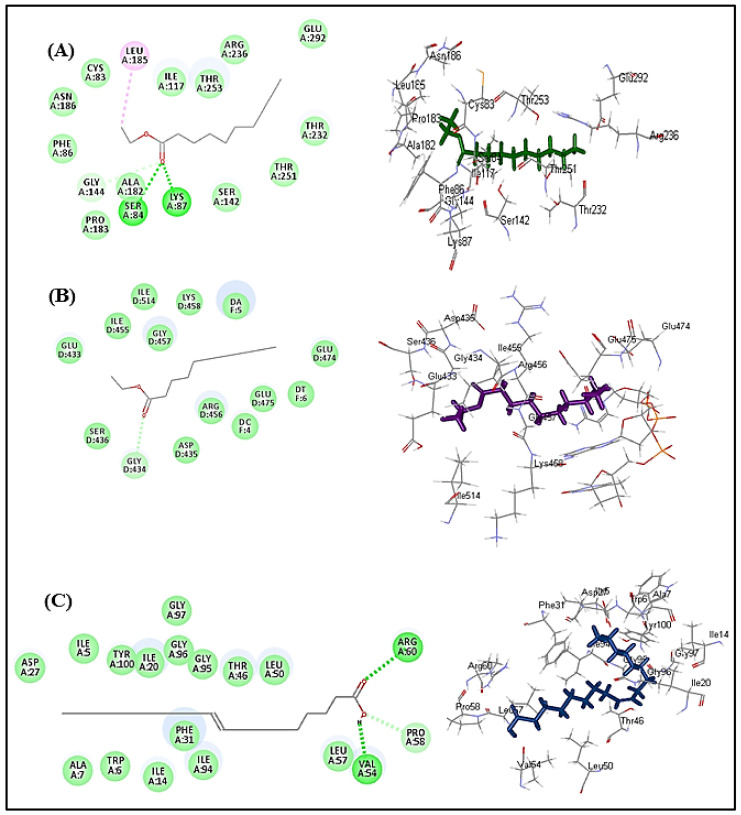
The 2D and 3D binding modes of decanoic acid ethyl ester within the binding sites of (**A**) *β*-lactamase, (**B**) DNA-gyrase and (**C**) oleic acid within the active sites of dihydrofolate reductase.

**Table 1 molecules-27-03744-t001:** LC-ESI-MS profiling of the methanol extract of the aerial parts of *L. libyca*.

Peak	RT	[M-H]^−^	Compound	Area (%)	Ref.
1	1.15	191	Quinic acid	12	[38]
2	7.24	917.4	Kaempferol 3-*O*-trihexoside 7-*O*-*α*-L-rhamnopyranoside	5	[6]
3	7.39	755.3	Kaempferol 3-*O*-dihexoside 7-*O*-*α*-L-rhamnopyranoside	24	[6]
4	7.99	739.3	Kaempferol 3-*O*-hexosyl-*α*- rhamnopyranoside-7-*O*-*α*-rhamnopyranoside	>1	[31]
5	8.51	569.4	Kaempferol 3*-O*-trihydroxybenzoyl- *α*-L-arabinofuranoside	12	[32]
6	8.84	977.4	Kaempferol 3-*O*-hexoside, 7-*O*-[4-hydroxy-3,5-dimethoxy-E-cinnamoyl dihexoside	2	[33]
7	10.12	961.3	Kaempferol 3-*O*-[4-Hydroxy-3,5-dimethoxy-E-cinnamoyl dihexoside 7*-O*-*α*-L-rhamnopyranoside	1	[34]
8	15.36	661.4	Apigenin 7-*O*-[3,4-di-*O*-acetyl-*α*-L-rhamnopyranosyl-hexoside	1	[35]
9	22.43	577.3	Kaempferol 3,7-di-*O*-*α*-rhamnopyranoside	1	[31]
10	24.70	555.3	Tetrahydroxyflavanone- trihydroxyflavone	1	[36]
11	27.65	281.3	Oleic acid	2	[39]
12	28.31	457.2	Hexahydroxyflavan, 3-*O*-Trihydroxybenzoyl	1	[37]
13	31.44	198.8	Decanoic acid ethyl ester	4	[40]

**Table 2 molecules-27-03744-t002:** Values of mean and standard deviations and coefficients of variation of chemical element mass fractions in the plant samples studied (mg/kg on a dry mass basis) (n = 3).

Elements	Mean ± SD * (mg/kg of DM)	Variability Coefficient (%)
Na	12,442.0 ± 1512.0	12.15
K	24,564.0 ± 2741.0	11.16
Ca	41,126.0 ± 6909.0	16.80
Mg	6651.0 ± 904.0	13.59
P	2864.0 ± 689.0	24.06
S	12,769.0 ± 1153.0	9.03
Fe	353.0 ± 52.0	14.73
Cu	8.13 ± 1.15	14.15
Zn	55.0 ± 6.6	12.00
Mn	38.00 ± 3.33	8.76
B	35.7 ± 4.6	12.89
Mo	3.14 ± 0.09	2.87
N	25,014.0 ± 2666.0	10.66

* SD: standard deviation (all values expressed on dry weight basis).

**Table 3 molecules-27-03744-t003:** Total phenolic and flavonoid contents of *L. libyca* (methanol extract).

Total Phenolic		Total Flavonoids	
(mg GAE/g Dry plant)	(mg GAE/g extract)	(mg quercetin/g Dry plant)	(mg quercetin/g extract)
11.49 ± 0.01	57.45 ± 0.04	7.10 ± 0.06	35.50 ± 0.30

**Table 4 molecules-27-03744-t004:** Comparison of total phenolic and flavonoid contents of *L. libyca* from Algeria (*L. libyca* A) and *L. libyca* Egyptian (*L. libyca* E) (methanol extract).

Plant	Total Phenolic Content mg GAE	Total Flavonoid Content mg Quercetin
*L. libyca* A/g Dry plant	11.49b ± 0.01	7.10a ± 0.06
*L. libyca* E/g fresh plant	25.26a ± 0.03	3.56b ± 0.102
LSD_0.05_	0.94	0.27

GAE: gallic acid equivalent*; L. libyca* A: *L. libyca* Algeria; *L. libyca* E: *L. libyca* Egypt. The same letters next to the numbers mean that there are no statistical differences between the examined objects.

**Table 5 molecules-27-03744-t005:** Cytotoxic activity of *L. libyca* against human cell lines.

Sample	LC_50_ (µg/mL)	LC_90_ (µg/mL)	Doxorubicin LC_50_ (μg/mL)	DMSO (100 ppm%)
HCT116	>100	>100	37.6	1
HepG 2	>100	>100	21.6	1

LC_50_: lethal concentration of the sample that causes the death of 50 % of the cells in 48 h. LC_90_: lethal concentration of the sample that causes the death of 90 % of the cells in 48 h. HePG 2: human hepatocellular carcinoma cell line. HCT116: colon cell line.

**Table 6 molecules-27-03744-t006:** Diameters of inhibition zones of bacterial strains treated with the methanol extract of *L. libyca* (n = 3).

Microorganisms	Diameter of Inhibition Zone (mm)		
	*L.**libyca* (10 mg/mL)	*L.**libyca* (20 µg/mL)	Amoxclav (30 µg/mL)
**Gram (+)**			
*Staphylococcus aureus* (*S. aureus*) (ATCC2592)	12 ± 0.25	NT	23 ± 0.15
*Listeria innocua* (*L. innocua*) (LIP74915)	11 ± 0.15	NT	
**Gram (−)**			
(ATCC700603)	12 ± 0.28	NT	22 ± 0.45
*Escherichia coli* (*E. coli*) (ATCC25922)	13 ± 0.17	NT	
**Yeast**			
*Candida albicans* (*C. albicans*) (IPA200)	NT	0.9 ± 0.18	23 ± 0.30

NT: not tested.

**Table 7 molecules-27-03744-t007:** Free binding energies (ΔG) in Kcal/mol of major compounds identified in *L. libyca* aerial parts within the active centers of specific enzymes implicated in the incidence of bacterial infections and resistance using molecular modeling.

Compounds	*β*-Lactamase	DNA-Gyrase	Dihydrofolate Reductase
Quinic acid (1)	−10.06	−7.50	−5.79
Kaempferol 3*-O*-trihydroxybenzoyl- *α*-L-arabinofuranoside (5)	−26.68	−7.95	−21.8055
Kaempferol 3,7-di-*O*-*α*-rhamnopyranoside (9)	0.18	16.68	5.44
Oleic acid (11)	−34.34	−25.68	−32.52
Decanoic acid ethyl ester (13)	−37.34	−27.16	−31.94
Cefuroxime	−61.76	ND	ND
Levofloxacin	ND	−9.90	ND
Trimethoprim	ND	ND	−30.10

## Data Availability

Not applicable.

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
