# Peer review of "Lobularia libyca: Phytochemical Profiling, Antioxidant and Antimicrobial Activity Using In Vitro and In Silico Studies"

_molecules, 2022, doi:10.3390/molecules27123744_

Round 1

Reviewer 1 Report

Basic methodology was not adhered for sample preparation of plant extracts. Using single solvent does not solve the purpose. Same principle must be follow for biological activity profile.

Reviewer 2 Report

Dear Authors,

The submission contains important results that should be of great interest to Molecules' readership. However, some imprecisions, corrections, and editing must be incorporated as major revisions to improve the manuscript. Since the authors did not insert line numbering, my suggestions/comments/queries were embedded within the attached file (authors' first submission). 

A general comment: Please note that the manuscript must be revised carefully for typo mistakes and English.  

Kind regards.   

Reviewer 3 Report

The papier „Phytochemical profiling of the aerial parts of the bioactive cooking herb Lobularia libyca and verification of its antioxidant and antimicrobial activity using in vitro and in silico studies” is complex, the molecular studies is very interesting. 

Round 2

Reviewer 1 Report

Please justify your answer Methanol : Water (8 : 2) as choice of your solvent with proper references. We can not carry the science based on your dreams. 

You may start your research with methanol or water but why methanol: water (8 : 2) ????

Why do you choose three set for antioxidant activity profile.

Reviewer 2 Report

Dear Authors, 

It seems that the manuscript has been improved. However, some comments were not taken into account nor answered. Please refer to my comments in the first round. Additionally, in order, for your manuscript to be improved, please see the revisions below:  

General comments:

A/English must be revised carefully, 

B/There are plenty of typo mistakes that must be checked,

C/Within the manuscript title: please change "antioxdant" to "antioxidant",

D/A rule to follow for the use of abbreviation: please define your abbreviation, at first mention, and then use it consecutively throughout your manuscript, 

Specific comments:

1/Please note that author that described species must not be in Italic: Boiss. Muschl (line 58 in the revised version). Please generalize the comment on the whole manuscript,

2/Line 66, please change to "Aromatic and medicinal plants are used ...",

3/Line 73, please change to "It consists of ...". In your sentence, "owing to" has no meaning and must be removed. Within the same line, please change "including endemic species 509" to "including 509 endemic species",   

4/Please check family name in line 78,

5/Line 81, please change "were" to "are",

6/Line 83, check family name,

7/Line 86, please change "because of" to "thanks to",

8/Line 91, scientific name must be carefully checked, 

9/Lines 91-98: paragraph must be rewritten to be more concise, and some details must (plant order, subclass, etc) should be removed. Line 98: please change "they grow.....sands" to "they grow on dry sandy soils",

10/ Line 134, please use "L libyca" instead of full name since the abbreviation is already defined in line 91. Please generalize the comment on the remaining text,

11/Line 142: please leave a space between value and unit "45 °C",

12/Line 145, please check for typos,

13/Line 186, it must not be a space between 2 and %: please correct to 2%,

14/Table 1: Please remove % for compounds 7 to 13, 

15/Table 2, please adjust the presentation of your values.

Kind regards. 

Round 3

Reviewer 1 Report

Please incorporate your justification statement in the discussion part or you may cite concerning reference into your methodology part.

Author Response

Dear Editor our responses are as follows:

Reviewer #1

Comment

Please incorporate your justification statement in the discussion part or you may cite concerning reference into your methodology part

Answer

Thanks for this remark from the reviewer; We have added some references about that.

Best regardes

Reviewer 2 Report

Dear Authors, 

The manuscript has been improved and therefore I recommend its publication in Molecules. 

Regards. 

Author Response

Dear Editor, we thank all the reviewers for their valuable inputs, which have given the manuscript a chance to reach a satisfactory level for publication.

(Added words correlated with the answers are marked in green in the revised manuscript)

Our responses are as follows:

Reviewer #2

Comments from the second reviewer

The manuscript has been improved and therefore I recommend its publication in Molecules.

Response

 We thank the Reviewer for this comment.

Thanks and Best Regards!

Yours Sincerely,
